# A Faster Maximum Cardinality Matching Algorithm with Applications in Machine Learning

**Nathaniel Lahn**[*]
School of Computing and Information Sciences
Radford University
Radford, VA 24142
nlahn@radford.edu

**Sharath Raghvendra**
Department of Computer Science
Virginia Tech Blacksburg, VA 24061
sharathr@vt.edu

**Jiacheng Ye**
Department of Computer Science
Virginia Tech Blacksburg, VA 24061
yjc0513@vt.edu

## Abstract

Maximum cardinality bipartite matching is an important graph optimization problem with several applications. For instance, maximum cardinality matching in a $\delta$-disc graph can be used in the computation of the bottleneck matching as well as the $\infty$-Wasserstein and the Lévy-Prokhorov distances between probability distributions. For any point sets $A, B \subset \mathbb{R}^2$, the $\delta$-disc graph is a bipartite graph formed by connecting every pair of points $(a, b) \in A \times B$ by an edge if the Euclidean distance between them is at most $\delta$. Using the classical Hopcroft-Karp algorithm, a maximum-cardinality matching on any $\delta$-disc graph can be found in $\tilde{O}(n^{3/2})$ time. [2] In this paper, we present a simplification of a recent algorithm (Lahn and Raghvendra, JoCG 2021) for the maximum cardinality matching problem and describe how a maximum cardinality matching in a $\delta$-disc graph can be computed asymptotically faster than $O(n^{3/2})$ time for any moderately dense point set. As applications, we show that if $A$ and $B$ are point sets drawn uniformly at random from a unit square, an exact bottleneck matching can be computed in $\tilde{O}(n^{4/3})$ time. On the other hand, experiments suggest that the Hopcroft-Karp algorithm seems to take roughly $\Theta(n^{3/2})$ time for this case. This translates to substantial improvements in execution time for larger inputs.

## 1   Introduction

Computing a maximum cardinality matching is a fundamental graph optimization problem. With origins in economics and logistics, matchings have found numerous applications. Computing popular distances between distributions, such as the Wasserstein distance as well as the Lévy-Prokhorov distance, can be reduced to a bipartite matching problem. In this paper, we consider the $\delta$-*disc graph matching* which is the following:

Given two sets $A$ and $B$ of $n$ two dimensional points and a parameter $\delta > 0$, a $\delta$-disc graph $G_\delta$ is a bipartite graph obtained by connecting any pair of vertices $(a, b) \in A \times B$ with an edge provided that the Euclidean distance between $a$ and $b$ is at most $\delta$, i.e., $\|a - b\| \leq \delta$. Let $m$ be the number of edges

---

[*]Authors are ordered by last name. All authors contributed equally to this work.

[2]We use $\tilde{O}(\cdot)$ to suppress poly-logarithmic terms in the complexity.

35th Conference on Neural Information Processing Systems (NeurIPS 2021).

in the graph $G_\delta$. A *matching $M$* is a set of vertex-disjoint edges in $G_\delta$. In the $\delta$-disc graph matching problem, we wish to compute a *maximum cardinality matching* in $G_\delta$, i.e., a matching that has the largest number of edges.

Any algorithm for computing a $\delta$-disc graph matching can also be used to compute a bottleneck matching as well as the Lévy-Prokhorov distance, both of which are defined next. Let $M \subseteq A \times B$ be any *perfect* matching of $A$ and $B$, which is a matching where every vertex of $A$ is matched, i.e., $|M| = n$. The edge of $M$ with the largest Euclidean length is its bottleneck edge. The *bottleneck matching* is a perfect matching $M^*$ whose bottleneck edge length is minimized. The Euclidean length of the bottleneck edge of $M^*$ is the *bottleneck distance* between $A$ and $B$.

**Distances between distributions:** Next, consider the case where $A$ and $B$ define discrete distributions $\mathcal{P}_A$ and $\mathcal{P}_B$ and let every point $a \in A$ (resp. $b \in B$) carry a probability of $1/n$, i.e., $\mathcal{P}_A(a) = 1/n$ (resp. $\mathcal{P}_B(b) = 1/n$). The $\infty$-Wasserstein distance between $\mathcal{P}_A$ and $\mathcal{P}_B$, denoted by $W_\infty(\mathcal{P}_A, \mathcal{P}_B)$, is simply the bottleneck distance between the point sets $A$ and $B$. The *Lévy-Prokhorov* distance between $\mathcal{P}_A$ and $\mathcal{P}_B$ is defined as follows: For $\varepsilon > 0$ and any subset $X \subseteq A$ (resp. $Y \subseteq B$), let $X^\varepsilon = \{b \in B \mid \exists a \in X \text{ such that } \|a - b\| \le \varepsilon\}$ (resp. $Y^\varepsilon = \{a \in A \mid \exists b \in Y \text{ such that } \|a - b\| \le \varepsilon\}$). Note that $X^\varepsilon \subseteq B$ and $Y^\varepsilon \subseteq A$. We say that the *Lévy-Prokhorov* [34] distance $\pi(\mathcal{P}_A, \mathcal{P}_B)$ is equal to the smallest value of $\varepsilon$ for which the following property is true:

$$\forall X \subseteq A, \mathcal{P}_A(X) \le \mathcal{P}_B(X^\varepsilon) + \varepsilon \text{ and, } \forall Y \subseteq B, \mathcal{P}_B(Y) \le \mathcal{P}_A(Y^\varepsilon) + \varepsilon;$$

here $\mathcal{P}_A(X) = \sum_{a \in X} \mathcal{P}_A(a) = |X|/n$ and $\mathcal{P}_B(Y) = \sum_{b \in Y} \mathcal{P}_B(b) = |Y|/n$. We can, therefore, write this condition as

$$\forall X \subseteq A, |X| \le |X^\varepsilon| + \varepsilon n \text{ and } \forall Y \subseteq B, |Y| \le |Y^\varepsilon| + \varepsilon n. \tag{1}$$

Determining if the exact bottleneck distance (equivalently $W_\infty(\mathcal{P}_A, \mathcal{P}_B)$) is $\le \delta$ can be done by simply finding the maximum cardinality matching $M$ in $G_\delta$. It is easy to see that $M$ is perfect if and only if the bottleneck distance is at most $\delta$. Similarly, by applying Hall's theorem, one can show that $\pi(\mathcal{P}_A, \mathcal{P}_B) \le \varepsilon$ if and only if the maximum cardinality matching $M$ in $G_\varepsilon$ has a size of at least $(1 - \varepsilon)n$. Thus, the $\delta$-disc graph matching directly relates to computing $\infty$-Wasserstein distance as well as the Lévy-Prokhorov distance between probability distributions.

**Computing $\delta$-disc graph matching:** One can use any of-the-shelf matching algorithm [18, 31, 33, 39] to compute a maximum cardinality matching in a $\delta$-disc graph. For instance, using the well-known Hopcroft-Karp algorithm will lead to an execution time of $O(m\sqrt{n})$ on any $\delta$-disc graph. The HK-Algorithm executes in phases. Each phase takes $O(m)$ time, and, in the worst-case, the algorithm converges to a maximum matching in $O(\sqrt{n})$ phases. The best-known exact algorithm for computing the exact bottleneck matching combines geometric data structures with the HK-algorithm in order to reduce the execution time of each phase from $O(m)$ to $\tilde{O}(n)$. As a result, they obtain an exact bottleneck matching in $\tilde{O}(n^{3/2})$ time.

Inspired by a series of algorithms for weighted matching in graphs with small separators [4, 27], Lahn and Raghvendra [28] presented a weighted approach to the maximum cardinality matching problem. This LR algorithm identifies a set of "edge separators" incident on $\omega$ "boundary vertices". These edges are assigned a weight of $1$ and all other edges receive a weight of $0$. A property satisfied by these separator vertices is that after their removal, every connected component in the graph has no more than $r$ vertices. Then, they present an algorithm to compute a perfect matching in $O(m\sqrt{r} + m\sqrt{\omega} + mr\omega/n \log n)$ time. As an application of their result, they show how to compute the bottleneck distance of any point sets $A$ and $B$ within a multiplicative factor of $(1 + \varepsilon)$ in $\tilde{O}(n^{4/3}\text{poly}(1/\varepsilon))$ time. The LR algorithm assigns dual weights to vertices and is similar in style to the Kuhn-Munkres [24] and Gabow-Tarjan [15] algorithms. The dual weights on vertices play a vital role in the proofs of correctness and efficiency of the LR algorithm.

In this paper, we make the following contributions:

- We remove the need to maintain dual weights in the LR algorithm, resulting in a significantly simpler algorithm.

- Using this algorithm, we show how to find a maximum cardinality matching in a unit-disc graph $G_\delta$ in time $\tilde{O}(n^{4/3}k^{1/3})$ where $k$ is the maximum number of points of $A \cup B$ contained in any disc of radius $\delta$. Note that our algorithm is asymptotically faster than the classical Hopcroft-Karp based algorithm when $k = o(\sqrt{n})$.

- Using our algorithm for $\delta$-disc graph matching, we show how to compute the exact bottleneck distance between point sets $A$ and $B$. When $\mathcal{P}_A$ and $\mathcal{P}_B$ are discrete distributions with each point having probability $1/n$, the bottleneck distance can be used to compute the distances $W_\infty(\mathcal{P}_A, \mathcal{P}_B)$) and $\pi(\mathcal{P}_A, \mathcal{P}_B)$. We are not aware of any previous polynomial time algorithms to compute the Lévy-Prokhorov distance. When $A, B$ are chosen uniformly at random from a unit square, our algorithm for the exact bottleneck distance runs in $\tilde{O}(n^{4/3})$ time. All previous algorithms take $\Omega(n^{3/2})$ time.

- We run experiments for the case where $A$ and $B$ are chosen uniformly at random from a unit square. Our experiments suggest that the Hopcroft-Karp algorithm on $G_\delta$ takes $\Theta(n^{3/2})$ time. In contrast, our algorithm runs substantially faster and executes in $\tilde{O}(n^{4/3})$ time.

Note that, for the HK-algorithm, the upper bound of $O(\sqrt{n})$ phases is only in the worst-case. As noted by Motwani [32], the Hopcroft-Karp algorithm converges, with high probability, to a maximum matching in $O(\log n)$ phases for expander graphs in general and Erdős-Rényi random graphs in particular. Similarly, does the HK algorithm execute asymptotically fewer than $\sqrt{n}$ iterations when used to compute bottleneck matchings? Interestingly, our experiments suggest that the answer to this question may be in the negative. Based on our experimental results, when $A$ and $B$ are drawn uniformly at random from a unit square and when $\delta$ is set to the bottleneck distance, the number of phases seem to grow at the rate of $\Omega(\sqrt{n})$. Therefore, bottleneck matching on random point sets may represent a natural hard instance for the HK-Algorithm. In this paper, we show how to overcome the $\Omega(n^{3/2})$ barrier for uniformly distributed point sets by developing an $\tilde{O}(n^{4/3})$ time algorithm.

For any $\varepsilon > 0$ and any point set $A \cup B$, the LR algorithm can also be used to compute a multiplicative $(1 + \varepsilon)$-approximation of the bottleneck matching in $\tilde{O}(n^{4/3}\mathrm{poly}(1/\varepsilon))$ time. This is done by using a grid where the side-length of each cell is a function of $\varepsilon$. The algorithm rounds every point to the closest cell center and finds a $\delta$-disc graph matching using the LR algorithm. See Section 6 of [28] for details. One can also use a similar approach to compute a multiplicative $(1 + \varepsilon)$ approximation of the Lévy-Prokhorov distance. Replacing the original LR algorithm with our dual-free implementation leads to simpler approximation algorithms.

**Applications:** Wasserstein distance has found numerous applications in machine learning and computer vision [3, 5, 8, 10, 14, 36]. Due to these applications, computing approximations of Wasserstein distances has received substantial attention [2, 9, 12, 26, 30, 35]. However, exact algorithms (even for discrete distributions) have a relatively high execution time [15, 24, 39]. The Lévy-Prokhorov distance have been extensively studied for its theoretical properties [11, 38]. For instance, Lévy-Prokhorov distance metrizes weak convergence on any separable metric space [19]. However, a brute-force algorithm based on the definition of this metric will require a search on exponentially many possible subsets causing it to seldom be used in practice [16]. However, we use Hall's theorem to show that the computation of the Lévy-Prokhorov metric reduces to the $\delta$-disc graph matching problem and so, it is only as hard as computing the $\infty$-Wasserstein distance, at least for discrete distributions of the type $\mathcal{P}_A$ and $\mathcal{P}_B$ described above.

In this paper, we provide exact algorithms for computing the $\infty$-Wasserstein and the Lévy-Prokhorov distances for 2-dimensional discrete distributions. Our algorithms can be useful in several scenarios. For instance, one can estimate Lévy-Prokhorov distances between any two fixed-dimensional continuous distributions by simply computing the distances between samples drawn from these distributions. From the fact that the Lévy-Prokhorov distance metrizes weak convergence, for large enough samples we can get accurate distance estimates. Faster high-precision algorithms are critical in obtaining such estimates; see [7, 6]. For high dimensional discrete distributions, Wasserstein distance is sometimes estimated by embedding them into a lower dimensional space and computing high precision solution in this space. For example, see the sliced Wasserstein distance [23].

In the emerging area of topological data analysis, high dimensional point clouds are characterized by two-dimensional point sets called persistence diagrams where each point represents the so-called birth and death times of a topological feature. Different high-dimensional point clouds can be compared by computing the bottleneck distance between the corresponding diagrams [17, 22, 1]. This has

led to development of practical implementations of bottleneck matching algorithms [17, 22]. More recently, other Wasserstein distances between persistence diagrams have also been considered. See for instance [25, 40].

The special case of computing bottleneck matching for point sets $A$ and $B$ that are drawn uniformly at random from a unit square has also received considerable attention. For instance, it has been used in the context of testing pseudo-random generators, average case analysis of bin packing algorithms [29], and also in statistics for analyzing the Glivenko-Cantelli convergence of empirical measures [37]. $\delta$-disc graphs have other applications as well, including in the modeling of the topology of ad-hoc wireless networks [20].

## 2 Matching algorithms

In this section, we present and compare two algorithms for solving the maximum cardinality matching problem on an arbitrary graph: The Hopcroft-Karp (HK) algorithm [18], and the LR algorithm [28]. In section 2.1, we introduce the basic definitions used by most combinatorial matching algorithms and give an overview of the HK algorithm. In section 2.2, we present the LR algorithm, highlighting the differences it has from the HK algorithm.

### 2.1  Preliminaries

Given any matching $M$, let $A_F$ and $B_F$ denote the vertices of $A$ and $B$ respectively that are not matched in $M$. We refer to these vertices as *free* vertices. An *alternating path* $P$ is a path that alternates between edges that are in the matching and those that are not in the matching. An *augmenting path* is an alternating path that starts and ends at a free vertex. We define the *length* of $P$ as the number of edges in $P$.

We can augment a matching $M$ along an augmenting path $P$ by updating the matching to $M \leftarrow M \oplus P$; where $\oplus$ denotes the symmetric difference operator. It is easy to see that augmenting a matching along an augmenting path $P$ increases the size of the matching $M$ by 1. Furthermore, it can be shown that $G$ has no augmenting paths with respect to a matching $M$ if and only if $M$ has maximum cardinality. These observations are the basis of the following commonly-used approach for computing a maximum-cardinality matching: repeatedly compute an augmenting path $P$ with respect to $M$ and augment $M$ along $P$ until $M$ has maximum cardinality. Since the largest possible matching has size at most $n$, any such algorithm will arrive at a maximum-cardinality matching after $n$ augmentations. This is the approach used by the classical Ford-Fulkerson and HK algorithms as well as the recent LR algorithm. However, the algorithms differ in the details of *how* these augmenting paths are found.

**Residual graph:** Given a matching $M$, the *residual graph* $G_M$ is a directed graph that assists in finding augmenting paths. The graph $G_M$ contains the same set of vertices $V$ as $G$. For any edge $(a, b)$ in $G$, if $(a, b) \in M$ then we add an edge directed from $a$ to $b$ to $G_M$. Otherwise, we add an edge directed from $b$ to $a$ to $G_M$. Furthermore, we create a source vertex $s$ and a sink vertex $t$ with the following additional edges. For each free vertex $b \in B_F$, we add an edge $(s, b)$ directed from the source $s$ to $b$ in $G_M$ and for each free vertex $a \in A_F$ we add an edge from $a$ to the sink $t$ in $G_M$. Note that, for the residual graph, we use $(u, v)$ to denote an edge directed from $u$ to $v$. On the other hand, for the undirected graph $G$, we may use $(u, v)$ and $(v, u)$ interchangeably to represent the same edge between vertices $u$ and $v$. Consider any directed path $P$ from $s$ to $t$ in the residual graph. Note that removing $s$ and $t$ from $P$ will result in an augmenting path. In the Ford-Fulkerson algorithm, a single augmenting path can be found in $O(m)$ time using any common graph search algorithm such as breadth-first search (BFS) or depth-first search (DFS), leading to an $O(mn)$ time algorithm. The HK and LR algorithms both improve upon this running time by finding potentially many augmenting paths in each iteration.

**HK algorithm:** Initially, let $M = \emptyset$. The HK algorithm executes in *phases*. A phase is divided up into two *stages*. The first stage executes a BFS starting from $s$ and identifies the length of the shortest path (path with the fewest edges) in $G_M$ from $s$ to every other vertex in $G_M$. Let $\ell_u$ be the length of the shortest path from $s$ to $u$ and let $\ell = \ell_t$. The algorithm then computes an *admissible graph* $\mathcal{A}$ consisting of all edges $(u, v)$ in $G_M$ such that (a) $\ell_u$ and $\ell_v$ are at most $\ell$ and (b) $\ell_v = \ell_u + 1$. Note that these edges capture the set of all minimum-length augmenting paths in $G_M$. The second

stage iteratively conducts multiple *partial DFSs* that start from $s$ and terminate early if a path to $t$ is found. Following the termination of a partial-DFS, all edges visited by it are removed from the residual graph. The algorithm proceeds to the next phase if a partial-DFS terminates without finding a path from $s$ to $t$. It can be shown that, in each phase, the HK algorithm finds a maximal set of vertex-disjoint shortest augmenting paths in $G_M$. Each phase involves execution of a single BFS and multiple partial DFSs. Since no two executions of DFS visit the same edge, the combined execution time of the multiple partial DFSs is bounded by $O(m)$.

Hopcroft and Karp showed that the length of the shortest augmenting path increases by at least 1 after each phase. Therefore, after $\sqrt{n}$ phases, the shortest augmenting path has length at least $\sqrt{n}$. Using this, they showed that there are no more than $\sqrt{n}$ free vertices remaining, all of which can be matched by augmenting along an additional $O(\sqrt{n})$ augmenting paths. Thus, the total number of phases executed by the algorithm is $O(\sqrt{n})$. Since each phase takes $O(m)$ time, the total time taken by the algorithm is $O(m\sqrt{n})$. Somewhat surprisingly, Hopcroft and Karp [18] also showed that the total length of all $n$ augmenting paths, across all phases, is only $O(n \log n)$.

## 2.2 A simplified implementation of the LR algorithm

Recently, Lahn and Raghvendra presented an algorithm [28] to compute a maximum cardinality matching. Their algorithm resembles the Kuhn-Munkres algorithm for weighted matching. In this section, we present a cleaner implementation of the LR algorithm. These simplifications result in a closer resemblance to the HK algorithm. Unlike the LR algorithm, our algorithm does not maintain any dual weights. In the following, we describe and contrast our algorithm with the HK-algorithm.

Apart from a bipartite graph $G(V, E)$, we are also given a subset $E_S \subseteq E$ of "separator edges" as input. For any separator edge $(u, v) \in E_S$, we denote the vertices $u$ and $v$ as *boundary vertices*. Let $B$ be the set of all boundary vertices. The analysis of the algorithm depends on $\omega = |B|$ and another parameter $r$ that is defined next. Consider the graph $G'(V, E \setminus E_S)$. Let $\mathbb{P} = \{\mathcal{P}_1, \ldots, \mathcal{P}_t\}$ be the set of connected components of $G'$ and let $\mathcal{V}_i$ and $\mathcal{E}_i$ be the set of vertices and edges of $\mathcal{P}_i$ for all $1 \leq i \leq t$. We refer to each $\mathcal{P}_i$ in $G'$ as a *piece* of the original graph $G$. Let $r = \max_{\mathcal{P}_i \in \mathbb{P}} |\mathcal{V}_i|$, i.e., the size of the piece of $G$ with the largest number of vertices.

**Setting weights on the edges of the graph $G$ and its residual graph $G_M$:** For any edge $(u, v) \in E$, we assign it a *weight* $w(u, v)$. For any separator edge $(u, v) \in E_S$, we set $w(u, v)$ to 1. For any other edge $(u', v') \in E \setminus E_S$, we set $w(u', v')$ to 0. Every edge $(u, v)$ in the residual graph inherits the weight of the corresponding edge $(u, v)$ in $G(V, E)$. All edges incident on the source $s$ and the sink $t$ in $G_M$ receive a weight of 0. For any path $P$, its weight is simply the sum of the weights of its edges.

**Preprocessing:** In the preprocessing step, our algorithm finds a maximum cardinality matching for each piece by applying the HK-Algorithm. Let $M$ be the union of these matchings computed across all pieces. At the end of this step, the difference $|M^*| - |M|$ is $O(\omega)$, where $M^*$ is the maximum cardinality matching in $G$. So, our algorithm has to find an additional $O(\omega)$ augmenting paths in order to compute a maximum cardinality matching.

The remaining $O(\omega)$ unmatched vertices are subsequently matched in *phases*. Like the HK algorithm, each phase of our algorithm consists of two *stages*. These stages somewhat resemble the stages of the HK algorithm. We highlight the differences in the description below.

**Stage 1:** In the first stage, our algorithm finds, for any vertex $v \in V$, the minimum weight path from $s$ to $v$ in the residual graph $G_M$ using the weights $w(\cdot, \cdot)$. Note that, since every edge weight is either 0 or 1, a standard BFS implementation can be modified to support such a minimum-weight search algorithm in $O(m)$ time by simply prioritizing edges of weight 0 over edges of weight 1. We call this modified version of BFS, *0/1 BFS*. For any vertex $v \in V$, let $\ell_v$ be the weight from $s$ to $v$ in $G_M$ as computed by the 0/1 BFS and let $\ell = \ell_t$. Any edge $(u, v)$ of $G_M$ is *admissible* if $\ell_u, \ell_v \leq \ell$ and $\ell_v = \ell_u + w(u, v)$. The *admissible* graph $\mathcal{A}$ is identical to $G_M$, except it contains only admissible edges. Similar to the HK algorithm, it can be shown that the admissible graph $\mathcal{A}$ captures every minimum-weight augmenting path.

**Stage 2:** The second stage of our algorithm finds a set of shortest augmenting paths (by weight). It does so by iteratively conducting partial-DFSs from $s$ until no augmenting path is found. Each partial-DFS immediately terminates if an augmenting path $P$ is found. Let $\mathcal{K}$ be the set of *affected pieces*, which are pieces that contain at least one edge of $P$. Unlike in the HK-Algorithm, the

matching $M$ is immediately augmented along $P$ and every edge visited by this partial-DFS that does not belong to an affected piece is deleted. Note that any edge from an affected piece that was visited by this partial-DFS does not get deleted and could be revisited by a later partial-DFS. In other words, edges in an affected piece can be visited multiple times within the same phase.

**Differences with HK algorithm:** The main differences between our algorithm and the HK algorithm are:

(1) Our algorithm assigns weights of 0 and 1 to the edges. No weights are assigned in the HK algorithm.

(2) Our algorithm has a preprocessing step that computes a maximum cardinality matching within each piece.

(3) In Stage 1, our algorithm executes a $0/1$-BFS instead of the BFS executed by HK algorithm.

(4) In Stage 2 of our algorithm, the partial-DFS reuses edges from affected pieces. As a result, in each phase, our algorithm may find augmenting paths that are not vertex-disjoint.

Differences (1) – (3) between HK algorithm and our algorithm do not impact the execution time of the algorithm by any more than a small constant factor. See Section F of the supplement for a discussion on this. The critical difference between the two algorithms is (4). Unlike the HK algorithm, Stage 2 of our algorithm reuses edges from affected pieces and computes a set of augmenting paths that are not necessarily vertex-disjoint. This allows for computing many more augmenting paths within each phase.

As we show later, the total number of phases executed as well as the augmenting paths computed by our algorithm are identical to those computed by the LR algorithm. Therefore, the analysis of Lahn and Raghvendra can be directly applied to our algorithm. They show that after each phase, the weight of the shortest augmenting path increases by at least one. After $\sqrt{\omega}$ phases, they show that there are $O(\sqrt{\omega})$ free vertices which can be matched using an additional $O(\sqrt{\omega})$ phases. Thus the total number of phases can be bounded by $O(\sqrt{\omega})$.

Edge revisits cause the execution time of a phase to increase. Note that an edge can be revisited only if it was inside an affected piece when it was most recently visited. Lahn and Raghvendra show that the total number of affected pieces is $O(\omega \log \omega)$ (a piece that is affected $k$ times is counted $k$ times in this sum). For graphs that admit recursive separators (such as planar and graphs with excluded minors), they show that the number of edges for any piece can be bounded by $O(mr/n)$ leading to an $O(\frac{mr\omega}{n} \log \omega)$ bound on the total number of revisits.

**Theorem 1.** *Consider a graph $G$ and a set of separator edges. Suppose each piece has at most $O(mr/n)$ edges. Our algorithm computes a maximum cardinality matching in $O(m\sqrt{r} + m\sqrt{\omega} + \frac{mr\omega}{n} \log n)$ time.*

For our algorithm, the assumption on an upper bound on the number of edges within each piece can be eliminated when graphs, such as those considered in this paper, support a dynamic data structure $\mathcal{D}$ of the following form: $\mathcal{D}$ can store any subset $A' \subseteq A$ of vertices and, given any query vertex $b \in B$, it can return a vertex $a \in A'$ that minimizes the weight of the edge $(b, a)$. Note that the weight of $(b, a)$ will be 1 only if every edge from $b$ to any vertex $a' \in A'$ has a weight of 1. If no edge exists between $b$ and any vertex of $A'$, then the data structure returns NULL. Suppose that $\mathcal{D}$ supports arbitrary insertions and deletions from $A'$, as well as queries, each in $\Phi(n)$ time. Then, one can use this data structure to dynamically maintain the set of unvisited nodes of $A$ during a 0/1 BFS or DFS. Consequently, one can execute 0/1 BFS and DFS in time $O(n\Phi(n))$. As a result, the execution time of our algorithm can be improved to $O(n\Phi(n)\sqrt{r} + n\Phi(n)\sqrt{\omega} + r\omega\Phi(n) \log n)$. In contrast, using $\mathcal{D}$ to execute a Hungarian Search inside the LR algorithm seems challenging.

**Theorem 2.** *Given a graph $G$ that supports a dynamic nearest neighbor data structure with query and update time of $\Phi(n)$, a maximum cardinality matching can be computed by our algorithm in $O(\Phi(n)(n\sqrt{r} + n\sqrt{\omega} + r\omega \log n))$. The HK algorithm computes a maximum cardinality matching in $O(n^{3/2}\Phi(n))$ time.*

**Equivalency to original LR algorithm:** The original algorithm maintains a dual weight $y(v)$ for every vertex $v \in A \cup B$ at any point during the algorithm. For any edge $(a, b) \in (A \times B) \cap E$, the dual weights satisfy the following:

$$y(b) - y(a) \leq w(a, b) \qquad \text{if } (a, b) \notin M, \tag{2}$$

$$y(a) - y(b) = w(a, b) \qquad \text{if } (a, b) \in M. \tag{3}$$

Additionally, their algorithm maintains the invariants that all free vertices of $B_F$ have the same dual weight of $y_{\max} = \max_{v \in A \cup B} y(v)$ and all free vertices of $A$ have the same dual weight of $0$. The *slack* $s(a, b)$ of any edge $(a, b) \in (A \times B) \cap E$ is defined as follows: if $(a, b) \notin M$, then $s(a, b) = w(a, b) - y(b) + y(a)$; otherwise, $(a, b) \in M$ and $s(a, b) = 0$.

In the original LR algorithm, the first stage adjusts the dual weights so that there is at least one zero-slack augmenting path in $G_M$. The second stage takes the subgraph consisting of zero slack edges and repeatedly executes a DFS from $s$. This DFS stops early if a path to $t$, i.e., an augmenting path, is found. After augmenting along a path, all edges visited by the DFS are deleted, unless they were in an affected piece.

Note that the only fundamental difference between the original version of the LR algorithm and our simplified version is the fact that we compute minimum-weight augmenting paths while they compute zero-slack augmenting paths. The following lemma, whose proof appears in Section A of the supplement, shows that the a zero slack path computed in the LR algorithm is also a minimum weight path. It follows that the two versions of the algorithm are equivalent.

**Lemma 1.** *During Stage 2 of the* LR *algorithm, an augmenting path has zero slack if and only if it has minimum weight.*

## 3 Applications

In this section, we show how our algorithm can be applied to efficiently compute a maximum cardinality matching on a $\delta$-disc graph, an optimal bottleneck matching, as well as the Lévy-Prokhorov distance between distributions. All applications considered are for point sets $A, B \subset \mathbb{R}^2$. For all applications, one can build a data structure $\mathcal{D}$ from Theorem 2 with $\Phi(n) = \log^{O(1)} n$ by using a dynamic Euclidean nearest neighbor data structure; see Section H.3 of the supplement for details

### 3.1 $\delta$-disc graph matching

Let $P = A \cup B$. Let $\mathbb{B}(p)$ be a ball centered at $p$ with radius $\delta$. Consider $k = \max_{p \in \mathbb{R}^2} |\mathbb{B}(p) \cap P|$, i.e., $k$ is the largest number of points of $A \cup B$ inside any ball of radius $\delta$. We refer to $k$ as the $\delta$*-density* of the point set $P$. We show that a maximum cardinality matching in a $\delta$-disc graph can be computed using our algorithm in $\tilde{O}(n^{4/3} k^{1/3})$ time. Thus, when the $\delta$-density $k = o(\sqrt{n})$, our algorithm outperforms the HK algorithm.

**Theorem 3.** *For any point set $P = A \cup B$ and a parameter $\delta > 0$, a maximum cardinality matching in the $\delta$-disc graph defined on $P$ can be computed in $\tilde{O}(n^{4/3} k^{1/3})$ time, where $k$ is the $\delta$-density of $P$.*

This result also extends to the case where the points of $A$ and $B$ are independently and identically distributed random variables drawn from distributions $\mathcal{P}_A$ and $\mathcal{P}_B$ respectively. We say that a distribution $\mathcal{P}$ has a $\delta$-density of $k$ if, for any ball $\mathbb{B}(p)$ of radius $\delta$, the probability that a point drawn from $\mathcal{P}$ lies inside the ball is at most $k/n$.

**Theorem 4.** *Let $A, B$ be drawn iid from distributions $\mathcal{P}_A$ and $\mathcal{P}_B$ respectively. For a parameter $\delta > 0$, a maximum cardinality matching in the $\delta$-disc graph defined on $A \cup B$ can be computed, with high probability, in $\tilde{O}(n^{4/3} k^{1/3})$ time, where $k$ is the maximum of the $\delta$-density of $\mathcal{P}_A$ and $\mathcal{P}_B$.*

**Proof of Theorem 3:** We show how a set of separator edges can be generated so that $\omega = O(n^{2/3} k^{2/3})$ and $r = O(n^{2/3}/k^{1/3})$. From Theorem 2 and since $n\sqrt{r} = O(n^{4/3}/k^{1/6})$, $n\sqrt{\omega} = O(n^{4/3} k^{1/3})$, and, $r\omega = O(n^{4/3} k^{1/3})$, the execution time of the LR algorithm can be bounded by $\tilde{O}(n^{4/3} k^{1/3})$. Next, we describe how to generate the separator edges $E_S$.

We use a *grid* $\mathbb{G}$ to generate the separator edges. Any grid consists of a set of equispaced horizontal and vertical lines that partition $\mathbb{R}^2$ into *cells*. Each cell is a square and any grid can be seen as a

set of these squares. Let $\mathcal{L}(\mathbb{G})$ denote the side-length of any cell $C \in \mathbb{G}$. We say that a cell $C$ is *non-empty* if $C \cap P \neq \emptyset$. Let $\theta = \lceil n^{1/3}/k^{2/3} \rceil$. To generate our pieces, we choose a grid $\mathbb{G}$ where the side-length of each cell is set to $\mathcal{L}(\mathbb{G}) = \theta\delta$. The separator edge set $E_S$ consists of all edges of the $\delta$-disc graph that have their endpoints in different cells. All such edges are assigned a weight of 1. Any edges whose endpoints are contained within the same cell of $\mathbb{G}$ are in $E \setminus E_S$ and are assigned a weight of 0. Any point that has at least one separator edge incident on it becomes a boundary vertex. To generate $\mathbb{G}$, we check $O(\theta)$ possible vertical and horizontal shifts and pick the one that minimizes the number of boundary vertices. We provide the details of generating $\mathbb{G}$ in Section B.1 of the supplement. Our choice of $\mathbb{G}$ guarantees that $\omega = O(n^{2/3}k^{2/3})$ for any point set, independent of its $\delta$-density. We show this in Section B.2 of the supplement.

**Bounding $r$:** By this definition, the number of vertices of any piece is bounded by the maximum number of points that can lie inside any cell of $\mathbb{G}$, i.e., $\max_{C \in \mathbb{G}} |C \cap P|$. Note that we can cover any cell of $\mathbb{G}$ with $\Theta(\theta^2)$ balls of radius $\delta$ each. Due to the $\delta$-density of $P$ being $k$, each of these balls can contain at most $k$ points and the total number of points inside any cell can be bounded by $O(\theta^2 k) = O(n^{2/3}/k^{1/3})$ as desired. In other words, $r = O(n^{2/3}/k^{1/3})$.

**Proof of Theorem 4:** The construction of the grid here will be identical to the one in the proof of Theorem 3. Note also that the bound on $\omega$ provided in that proof depends only on the construction of $\mathbb{G}$ and not on the $\delta$-density of the point set. Therefore, the same bound continues to hold here as well. In Section C of the supplement, we use the $\delta$-density of $\mathcal{P}_A$ and $\mathcal{P}_B$ along with Chernoff's bound to prove that $r = O(n^{2/3}/k^{1/3})$ points with high probability.

## 3.2 Bottleneck distance

In this section, we assume that $A$ and $B$ are points drawn uniformly at random from a unit square. From the work of Leighton and Shor [29], we know that, for appropriate constants $c_{\min}$ and $c_{\max}$, the optimal bottleneck distance is at least $\delta_{\min} = \frac{c_{\min} \log^{3/4}(n)}{\sqrt{n}}$ and at most $\delta_{\max} = \frac{c_{\max} \log^{3/4}(n)}{\sqrt{n}}$ with very high probability (probability exceeding $1 - 1/n^\alpha$ for some $\alpha = \Omega(\sqrt{\log n})$). Observe that the optimal bottleneck distance will be equal to the length of some edge of $A \times B$. As in the work of Efrat *et al.* [13], our algorithm will use a selection algorithm of Katz and Sharir [21] to find the $j$th smallest edge, $1 \leq j \leq n^2$ in $O(n^{4/3} \log^2 n)$ time. Let $d(j)$ be the length of the $j$th smallest edge returned by their algorithm. This allows us to execute a binary search over the edges of $A \times B$, ordered by their length. Let $g_{\min} = 1$ and $g_{\max} = n^2$. We repeat the following process until $g_{\max} = g_{\min} + 1$: We choose $j = \lfloor (g_{\max} + g_{\min})/2 \rfloor$ and find the $j$th smallest edge whose length is denoted by $d(j)$. If $d(j) \geq \delta_{\max}$, we set $g_{\max} \leftarrow j$. If $d(j) \leq \delta_{\min}$, we set $g_{\min} \leftarrow j$. Otherwise, $\delta_{\min} \leq d(j) \leq \delta_{\max}$, and we find the maximum cardinality matching in a $\delta$-disc graph where $\delta$ is set to $d(j)$. If we obtain a perfect matching, we set $g_{\max} = j$. Otherwise, the maximum matching is not perfect, and we set $g_{\min} = j$. When the algorithm terminates, $d(g_{\max})$ is the optimal bottleneck distance.

**Analysis:** The algorithm makes $O(\log n)$ many guesses. These guesses are found by a selection algorithm that runs in $O(n^{4/3} \log^2 n)$ time. For each guess $j$ where $\delta_{\min} \leq d(j) \leq \delta_{\max}$, we must compute a maximum-cardinality matching on a $\delta$-disc graph, which takes $O(n^{4/3}k^{1/3})$ time using the LR algorithm. Since, $\mathcal{P}(A)$ and $\mathcal{P}(B)$ are the uniform distribution, their $\delta$-density increases as $\delta$ increases. Therefore, the $\delta_{\max}$-density of $\mathcal{P}(A)$ and $\mathcal{P}(B)$ will be an upperbound on the $\delta$-density for any execution of the LR algorithm. The probability that any random point lies within any ball of radius $\delta_{\max}$ is at most $(2\delta_{\max})^2$, which is $\Theta(\log^{3/2}(n)/n)$. Therefore, the $\delta_{\max}$-density of $\mathcal{P}(A)$ and $\mathcal{P}(B)$ is at most $k = O(\log^{3/2}(n))$. Applying Theorem 4 gives the following:

**Theorem 5.** *Let $A, B$ be drawn uniformly at random from a unit square. An optimal bottleneck matching can be computed between $A$ and $B$, with high probability, in $\tilde{O}(n^{4/3})$ time.*

**Practical considerations:** The algorithm described uses two black-boxes that are impractical and have hidden high constants in the Big-O notation. (a) the algorithm relies on a dynamic nearest neighbor data structure, and, (b) the algorithm uses the selection algorithm of Katz and Sharir [21]. In Section D of the supplement, we address both (a) and (b) by presenting more practical alternatives.

### 3.3 Lévy-Prokhorov distance

In this section, we show that we can use our algorithm for the $\delta$-disc graph matching to also compute the Lévy-Prokhorov distance.

We describe a simple algorithm to decide if $\pi(\mathcal{P}_A, \mathcal{P}_B)$ greater than or at most $\varepsilon$. We compute a maximum cardinality matching $M$ in an $\varepsilon$-disc graph. Let $A_F$ and $B_F$ be the free vertices with respect to $M$. Then, we say that the distance is greater than $\varepsilon$ if $|A_F| > \varepsilon n$. Otherwise, we say that the distance is at most $\varepsilon$. Using a binary search similar to the one described in Section 3.2, we can determine the distance in $\tilde{O}(n^{4/3} k^{1/3})$ time.

**Proof via Hall's theorem:** Given any bipartite graph $G(A \cup B, E)$, for any set $X \subseteq A$, the neighborhood $\mathcal{N}(X)$ is the set of all vertices of $B$ that share an edge with at least one vertex of $X$. Thus, $X^\varepsilon$ is the neighborhood of $X$ in an $\varepsilon$-disc graph. The deficiency of a graph with respect to $A$ is $\mu(A) = \max_{X \subseteq A} |X| - \mathcal{N}(X)$. Hall's theorem says that a bipartite graph has a perfect matching if and only if the deficiency of the graph with respect to $A$ is non-positive. Hall's theorem can be generalized to the following.

**Lemma 2.** *For any bipartite graph $G(A \cup B, E)$, where $|A| = |B| = n$, and for any integer $k > 0$, the deficiency with respect to $A$ or $B$ is $k$ if and only if the maximum cardinality matching is of size $n - k$.*

The proof of this generalization follows in a straight-forward way from the Hall's theorem. For the sake of completion, we provide this proof in Section E of the supplement. Next, we show that the algorithm described here correctly computes the Lévy-Prokhorov distance.

Recollect that, our algorithm returns the distance to be greater than $\varepsilon$ if $|A_F| > \varepsilon n$. By Lemma 2, we conclude that the deficiency of the graph is greater than $\varepsilon n$, i.e., there is a set $X \subseteq A$ such that $|X| - |X^\varepsilon| > \varepsilon n$. Thus, Equation 1 does not hold and the distance is greater than $\varepsilon$.

Our algorithm returns a distance at most $\varepsilon$ if $|A_F| \leq \varepsilon n$. In this case, from Lemma 2, the deficiency of the graph with respect to $A$ is less than $\varepsilon n$, i.e., for every subset $X \subseteq A$, $|X| - |X^\varepsilon| \leq \varepsilon n$. Note that $|A_F| = |B_F| \leq \varepsilon n$ and so an identical argument applies for $B$ as well. Thus, Equation 1 holds and the distance is at most $\varepsilon$. We conclude that the algorithm terminates with the correct $\varepsilon$.

**Theorem 6.** *Let the point sets $A$ and $B$, $|A| = |B| = n$ describe two distributions $\mathcal{P}_A$ and $\mathcal{P}_B$ where each point has a probability of $1/n$ associated with it. The Lévy-Prokhorov distance $\pi(\mathcal{P}_A, \mathcal{P}_B)$ can be computed in $\tilde{O}(n^{4/3} k^{1/3})$ time where $k$ is the $\delta$-density of $A \cup B$.*

## 4 Experimental results

In this section, we compare the performance of the HK and LR algorithms when applied to computing an exact bottleneck matching between equal-sized point sets $A, B \subset \mathbb{R}^2$ drawn uniformly at random from a unit square, where $n = |A| + |B|$.

**Experimental setup:** For each value of $n$ in $\{100, 1000, 5000, 10000, 50000, 100000, 500000, 1000000, 1500000\}$, we execute 10 runs. For each run, we uniformly sample points from a unit square to obtain the point sets $A$ and $B$. Next, we compute a bottleneck matching between $A$ and $B$ separately, using both the HK algorithm and our algorithm, and record performance metrics for both algorithms. We execute our experiments on a server running CentOS Linux 7, with 12 Intel E5-2683v4 cores and 128GB of RAM.[3]

When guessing the bottleneck distance for each run, instead of enforcing that the number of guesses is $O(\log n)$ it is sufficient in practice to continue the binary search on $\delta$ until the relative error becomes less than a sufficiently small value $\varepsilon$ (see Section D of the supplement). Both the HK-based algorithm and the LR-based algorithm use the same strategy for guessing the bottleneck distance in the experiments.

**Experimental results:** For each figure, the data presented for each value of $n$ is averaged over all 10 runs. Error bars represent a single standard deviation. Figure 1 presents the actual running time of both algorithms, summed over all guesses of the bottleneck distance. For datasets with more than $10^6$ points, our algorithm takes roughly half as much time as the HK algorithm and the gap seems

---

[3]Our implementations of our algorithm and HK can be found at https://github.com/nathaniellahn/JOCGV3

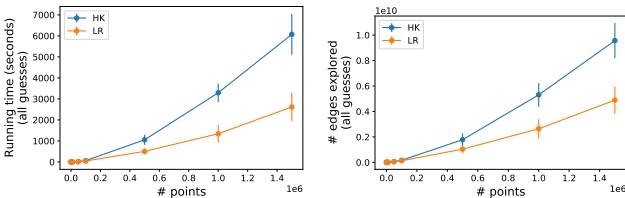

Figure 1: A running time comparison between the HK algorithm and our algorithm. Left: Comparison of actual running time. Right: Comparison of total number of edge visits.

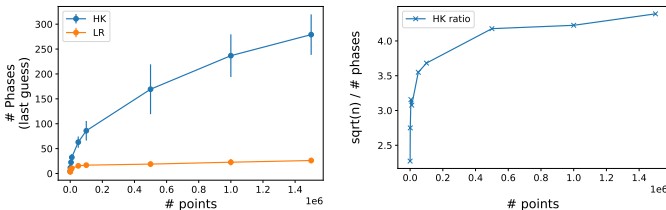

Figure 2: Data for number of phases for the final bottleneck guess. Left: A comparison of the number of phases for the HK algorithm and our algorithm. Right: $\sqrt{n}$ divided by the number of phases executed by the HK algorithm.

to grow as the input size increases. However, the actual running times can be affected by several factors including the exact implementation details and execution environment. Therefore, we focus on comparing metrics that are accurate independent of the exact implementation details. Recall that both algorithms combine variants of BFS and DFS to compute augmenting paths. As a result, the total number times edges are visited during each algorithm acts as an implementation-independent proxy of the running time. Figure 1 shows the total number of edge visits for both algorithms. Note that this data seems to follow a similar trend to the actual running times of the algorithms.

Next, we summarize our observations that help account for this difference in performance of the two algorithms. For more details, see Section G of the supplement. Recall that there are four main differences (1) – (4) between the HK algorithm and our algorithm. As discussed in Section F of the supplement, differences (1) – (3) do not have any direct significant impact on the relative running times of the two algorithms; the most significant difference is (4) – Stage 2 of the our algorithm reuses edges from affected pieces. This reuse of edges has two main effects on the efficiency of the our algorithm. First, we find that our algorithm executes significantly fewer phases than the HK algorithm. Specifically, as the guess of the bottleneck distance approaches the actual bottleneck distance, our results suggest that the number of phases executed by the HK algorithm seems to grow at a rate of $\Theta(\sqrt{n})$ – exhibiting its worst-case analysis. In contrast, the number of phases executed by our algorithm grows at a much slower rate (see Figure 2). This explains why the our algorithm runs faster than the HK algorithm. The second impact of allowing for edge revisits is that a single edge can be revisited, perhaps many times, during a single phase. Despite this, the total number of edges visited by our algorithm is still significantly less than the total number of edges visited by the HK algorithm (see Figure 1).

## 5 Conclusion

We consider the maximum cardinality matching problem and present a simplification of a recent algorithm by Lahn and Raghvendra [28]. In particular, we eliminate the need to maintain dual weights in their algorithm. This not only leads to a simpler algorithm but also results in new and improved exact algorithms for computing the $\delta$-disc graph matching, bottleneck matching, as well as the $\infty$-Wasserstein and the Lévy-Prokhorov distances, in low-density settings. We would like to conclude by stating the following open question: Can we design a parallel combinatorial algorithm to compute a $\delta$-disc graph matching?

**Acknowledgements** We would like to acknowledge, Advanced Research Computing (ARC) at Virginia Tech, which provided us with the computational resources used to run the experiments. Research presented in this paper was funded by NSF CCF-1909171. We would like to thank the anonymous reviewers for their useful feedback.

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
