## A   Proof of Lemma 1

During Stage 2 of LR algorithm, an augmenting path has zero slack if and only if it has minimum weight.

*Proof.* Let $M, y(\cdot)$ be matching and set of dual weights that are feasible. At any point during the original version of the LR algorithm, consider any augmenting path $P$ in $G_M$ whose first vertex is $b'$ and whose last vertex is $a'$. Then, it follows that,

$$\sum_{(u,v)\in P} w(u,v) = \sum_{(b,a)\in P\setminus M} (y(b) - y(a) + s(b,a)) + \sum_{(a,b)\in P\cap M} (y(a) - y(b))$$

$$= y(b') - y(a') + \sum_{(u,v)\in P} s(u,v)$$

$$= y_{\max} + \sum_{(u,v)\in P} s(u,v).$$

The last equality follows from the invariant that all free vertices of $B_F$ have the same dual weight $y_{\max}$ and all vertices of $A_F$ have a dual weight of $0$. Since all slacks are non-negative, an augmenting path has minimum weight if and only if it has zero slack. $\qquad\square$

## B   Omitted details for proof of Theorem 3

### B.1   Grid generation

Next, we describe how to generate the grid $\mathbb{G}$. Without loss of generality, we assume that the points are contained inside a bounding box whose bottom left corner has coordinates $(\theta\delta, \theta\delta)$ and whose top right corner has coordinates $(\Delta, \Delta)$, where $\Delta$ is of the form $t\theta\delta$ for some integer value $t$, i.e., when we choose $\mathbb{G}$, the number of vertical and horizontal lines will be $t$ each.

Consider another grid $\mathbb{G}_\delta$ where each cell $C \in \mathbb{G}_\delta$ has a side-length of $\delta$. Let $\mathbb{Y} = \{y_1, \dots, y_t\}$ be the set of vertical lines that define $\mathbb{G}_\delta$. For $1 < i < t$, let $\mu_i$ be the number of points of $P$ that lie between the vertical lines $y_{i-1}$ and $y_{i+1}$; $\mu_1$ and $\mu_t$ are defined to be $0$. For any subset $Y \subset \mathbb{Y}$, let $\mu(Y)$ denote $\sum_{y_i \in Y} \mu_i$. We generate $\theta$ different candidates for the vertical lines of $\mathbb{G}$ as follows: For $0 \le i < \theta$, let $\mathbb{Y}_i$ be a set of vertical lines that are spaced $\theta\delta$ apart. The left-most line is given by $x = i\delta$ and $\mathbb{Y}_i = \{x = i\delta + j\theta\delta \mid 0 \le j < t - 1\}$. In other words, the set $\mathbb{Y}$ is partitioned into $\theta$ groups. We pick the set $\mathbb{Y}_i$ that minimizes $\mu(\mathbb{Y}_i)$. This can be done in $O(n \log n)$ time. A symmetric construction also applies for choosing the set of horizontal lines. $\mathbb{G}$ is the grid resulting from the selected horizontal and vertical lines.

### B.2   Bound on $\omega$

**Bounding $\omega$:** We will show that $\omega = O(n^{2/3}k^{2/3})$. For any vertical line $y_i \in \mathbb{Y}$, any edge of length at most $\delta$ that crosses $y_i$ will have its end points between $y_{i-1}$ and $y_{i+1}$. Therefore, if $y_i$ is chosen as a vertical line for $\mathbb{G}$, then $\mu_i$ will be an upper bound on the number of boundary vertices whose edges cross $y_i$. Thus, if $\mathbb{Y}_i$ is chosen as the set of vertical lines of $\mathbb{G}$, then $\mu(\mathbb{Y}_i)$ will be an upper bound on the number of boundary vertices created in $\mathbb{G}$ due to the vertical lines. For any point $p$, let $y_i$ and $y_{i+1}$ be the vertical lines between which $p$ lies. Then $p$ contributes to $\mu_i$ and $\mu_{i+1}$. Thus, $\sum_{y_i \in \mathbb{Y}} \mu_i \le 2n$. Since each $y_j \in \mathbb{Y}$ participates in exactly one subset $\mathbb{Y}_i$, $\min_{0 \le i < \theta} \mu(\mathbb{Y}_i) \le (1/\theta)\sum_{0 \le i < \theta} \mu(\mathbb{Y}_i) = (1/\theta)\sum_{y_j \in \mathbb{Y}} \mu_j \le 2n/\theta$. Since $\theta = \lceil n^{1/3}/k^{2/3} \rceil$, we get a bound on the total number of boundary vertices formed due to the vertical lines to be $O(n^{2/3}k^{2/3})$. An identical argument bounds the number of boundary vertices created due to the horizontal lines to be $O(n^{2/3}k^{2/3})$ leading to $\omega = O(n^{2/3}k^{2/3})$ as desired.

## C   Bound on $r$ in proof of Theorem 4

We bound the number of points of $A$ in any piece by $O(n^{2/3}/k^{1/3})$. An identical argument also applies for $B$. Let $\mathbb{C} \subseteq \mathbb{G}$ be the subset of cells of $\mathbb{G}$ that contain a non-zero probability with respect

to distribution $\mathcal{P}_A$. Since the $\delta$-density of $\mathcal{P}_A$ is $k$, and any cell of $\mathbb{G}$ can be covered by $O(\theta^2)$ balls of radius $\delta$, the total probability within any cell is $O(1/(k^{1/3}n^{1/3}))$. We partition the cells in $\mathbb{C}$ into $\Theta(n^{1/3}k^{1/3})$ groups where each group has a total probability density of $\Theta(1/(n^{1/3}k^{1/3}))$. For each group of cells, we show that with high probability that it contains at most $\Theta(n^{2/3}/k^{1/3})$ points. Then, we apply union bound on the $\Theta(n^{1/3}k^{1/3})$ groups to show that no group contains more than $cn^{2/3}/k^{1/3}$ points, for some constant $c$.

Fix a group of cells $C$. Recollect that $C \subset \mathbb{C}$ and the total probability density for all cells of $C$ combined in $\Theta(1/(n^{1/3}k^{1/3})) = c'/(n^{1/3}k^{1/3})$ for some constant $c' > 0$. Now define a random variable $X$ to be 1 if a point $p$ chosen from the distribution $\mathcal{P}_A$ is inside one of the cells in the group $C$. Otherwise, if $p$ is not contained in any of the cells from the group $C$, then $X$ is 0. Note that $X$ is a Bernoulli random variable with $p = c'/n^{1/3}k^{1/3}$. Now, consider the $n$ points chosen independently from $\mathcal{P}_A$. Let $X_i$ be the outcome of the $i$th point. Then, $Y = \sum_{i=1}^{n} X_i$ would be the number of points of $A$ inside any of the cells of the group $C$.

Since $\mathbb{E}[X_i] = c'/(k^{1/3}n^{1/3}))$ and from linearity of expectation, we conclude that $\mathbb{E}[Y] = \mathbb{E}[\sum_{i=1}^{n} X_i] = \sum_{i=1}^{n} \mathbb{E}[X_i] = c'n^{2/3}/k^{1/3}$. Applying Chernoff's bound,

$$\Pr[Y > 2c'n^{2/3}/k^{1/3}] \leq e^{-c'n^{2/3}/k^{1/3}}.$$

Thus, by applying union bounds on the $\Theta(n^{1/3}k^{1/3})$ groups, we get that with probability at least $1 - (\frac{n^{1/3}k^{1/3}}{e^{n^{2/3}/k^{1/3}}})$ no cell of the grid $\mathbb{G}$ contains more than $2c'n^{2/3}/k^{1/3}$ many points.

## D  Practical considerations

Recall that the algorithm of Section 3.2 uses two black-boxes that are impractical and have hidden high constants in the Big-O notation. (a) the algorithm relies on a dynamic nearest neighbor data structure, and, (b) the algorithm uses the selection algorithm of Katz and Sharir [21]. In order to address (a), we observe that each vertex has an expected degree of $O(k) = O(\log^{3/2}(n))$. Therefore, it is acceptable to explicitly compute the roughly $m = O(n\log^{3/2}(n))$ edges of the $\delta$-disc graph and apply standard $O(m)$ time graph search algorithms.

We use the following approach to explicitly construct the edges of the $\delta$-disc graph: Let $\hat{\mathbb{G}}$ be a arbitrarily-placed grid with each cell having side-length $\delta$. For any point $a \in A$, let $N(a)$ be the set of all cells of $\hat{\mathbb{G}}$ whose boundary is within a distance of $\delta$ from $a$. Note that $N(a)$ contains exactly 9 cells – the cell containing $a$ itself along with its 8 adjacent neighbors. Any point $b$ within a distance $\delta$ of $a$ must lie within one of the cells of $N(a)$. Therefore, it is sufficient to enumerate, for every cell $\square \in N(a)$, every point $b \in B \cap \square$, and add $(a, b)$ to the $\delta$-disc graph if and only if the distance between $a$ and $b$ is at most $\delta$.

To implement this algorithm for edge-generation, we must be able to compute a bidirectional mapping between an input point and the grid cell that contains the point. To facilitate this mapping, we use the following approach: First, sort the points of $A \cup B$ by their $x$-coordinates. Each non-empty column in the grid corresponds to a contiguous interval of points in this $x$-sorted list. For each such column, sort the points within the corresponding $x$-sorted interval by their $y$-coordinates. Next further divide each column interval of the $x$-sorted list into non-empty rows by further splitting the $x$-sorted intervals into $y$-sorted intervals (within each column). Given any cell $\square$, the points contained within $\square$ can be found by first binary searching through the $x$-sorted intervals in order to find the list of points within the column of $\square$, and then binary searching through the $y$-sorted intervals of this column in order to find the corresponding interval for the row of $\square$. Thus, given the bounds of a cell $\square \in \hat{\mathbb{G}}$, the points of $(A \cup B) \cap \square$ can be found in $O(\log n)$ time. Furthermore, as these intervals are being constructed, the algorithm stores, along with each point, the boundary region of grid cell that contains that point. Thus, a point can be mapped to its containing grid cell in $O(1)$ time. Note that the set of intervals can be computed using $O(n)$ space (since empty rows and columns need not have a corresponding interval) and $O(n\log n)$ time. As a result, the total time spent constructing the $\delta$-disc graph is $O(n\log n + \sum_{a \in A}\sum_{\square \in N(a)}|B \cap \square|)$. Since the $\delta$-disc graph has a $\delta$-density of $O(\log^{3/2})$ with high probability, and the cells of $N(a)$ can be covered by $O(1)$ discs of radius $\delta$, the $\delta$-disc graph creation algorithm runs in $\tilde{O}(n)$ time.

In order to address (b), instead of executing an integer binary search over the $n^2$ edges of $A \times B$, we simply execute a binary search over the interval $[\delta_{\min}, \delta_{\max}]$. Initially, we set $g_{\min} \leftarrow \delta_{\min}$ and $g_{\max} \leftarrow \delta_{\max}$. We repeat the following process until $\delta_{\max} \leq \delta_{\min} + \varepsilon$ for a sufficiently small error parameter $\varepsilon$. For a guess $\delta = (g_{\max} + g_{\min})/2$, we compute a maximum-cardinality matching on the $\delta$-disc graph. If the result is a perfect matching, we set $g_{\max} \leftarrow \delta$. Otherwise, we set $g_{\min} \leftarrow \delta$. This algorithm terminates after $O(\log \frac{1}{\varepsilon})$ iterations, where each iteration executes our algorithm. Under the reasonable assumption that $\varepsilon = 1/n^{O(1)}$, the number of iterations remains only $O(\log n)$.

# E   Proof of Lemma 2

Recollect that for any bipartite graph $G(A \cup B, E)$ and for any set $X \subseteq A$, the neighborhood $\mathcal{N}(X)$ is the set of all vertices of $B$ that share an edge with at least one vertex of $X$. The deficiency of a graph with respect to $A$ is $\mu(A) = \max_{X \subseteq A} |X| - \mathcal{N}(X)$. Hall's theorem says that a maximum cardinality matching in a bipartite graph matches every vertex of $A$ if and only if the deficiency of the graph with respect to $A$ is non-positive. Using this, we will show the following lemma.

**Lemma:** For any bipartite graph $G(A \cup B, E)$, where $|A| = |B| = n$, and for any integer $k > 0$, the deficiency with respect to $A$ or $B$ is $k$ if and only if the maximum cardinality matching is of size $n - k$.

**Proof:** We begin by proving (i) and (ii) and then use it to prove the lemma.

(i) Suppose the deficiency of the graph with respect to $A$ is $k$, then we will show that any matching has a cardinality at most $n - k$. From the definition of deficiency, there is a subset $X \subseteq A$ such that $|X| - |\mathcal{N}(X)| = k$. In any maximum cardinality matching $M$, the vertices of $X$ can only match to a vertex in the neighborhood $\mathcal{N}(X)$. Since, there are $k$ fewer neighbors, at least $k$ vertices in $X$ will remain unmatched in $M$, i.e., $|M| \leq n - k$.

(ii) If the maximum cardinality matching $M$ is of size $n - k$, then we will show that the deficiency of the graph is at most $k$. We add $k$ dummy vertices to the set $B$ and connect them to every vertex of $A$. As a result, the new graph will now match every vertex of $A$: every vertex of $A$ that was unmatched with respect to $M$ will now match to a dummy vertex of $B$. By Hall's theorem, the deficiency of this graph with respect to $A$ is $0$. Note, however, that the dummy vertices raised the deficiency of every subset $X \subset A$ by exactly $k$. Therefore, the deficiency of the graph prior to adding the dummy vertices is at most $k$.

Suppose the deficiency of a graph is $k$. From (i), the maximum cardinality matching is of size $\leq n - k$, say $n - k'$ for some $k' \geq k$. Since the maximum cardinality matching is of size $n - k'$, from (ii), we conclude that the deficiency of the graph is at most $k'$, i.e., $k \leq k'$. Since both $k \geq k'$ and $k \leq k'$, we get $k = k'$ and the maximum cardinality matching is of size exactly $n - k$.

# F   Details of differences between the HK algorithm and our algorithm

Recall that the HK algorithm and our algorithm differ in the following three ways: (1) The LR algorithm assigns weights of 0 and 1 to the edges. No weights are assigned in the HK algorithm. (2) The LR has a preprocessing step that computes the maximum cardinality matching for each piece. (3) In Stage 1, the LR algorithm executes a $0/1$-BFS instead of the BFS executed by HK algorithm. Next, we explain why these three differences do not have any significant effect on the comparative running times of the two algorithms. In (1), for any edge $(u, v)$, one can determine whether $(u, v)$ is in $E_S$ in $O(1)$ time. In (2), the LR algorithm executes the HK algorithm for each piece each of which contains at most $r$ vertices. The time taken across all pieces is $\sum_{i=1}^{t} O(|\mathcal{E}_i|\sqrt{|\mathcal{V}_i|}) = O(m\sqrt{r})$. In (3), the 0/1 BFS executed in Stage 1 of the algorithm is almost identical to the BFS executed by the HK algorithm, except that it prioritizes weight 0 edges before the weight 1 edges. It can be implemented to take $O(m)$ time.

# G   Additional experimental results

Recall that there are four main differences between the HK algorithm and our algorithm. Our algorithm must incorporate weights of 0 and 1 into the edges. For arbitrary inputs, the $\delta$-disc graph

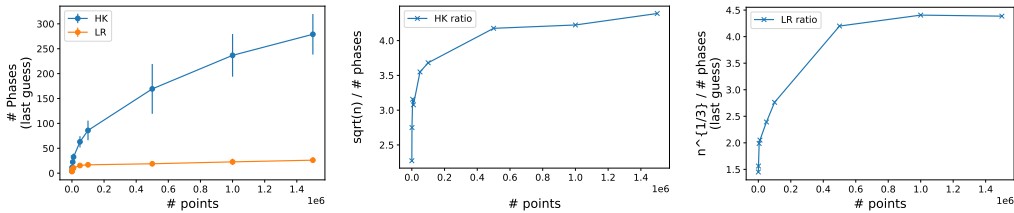

Figure 3: Comparisons of number of phases for the final bottleneck guess. Left: A comparison of the number of phases for the HK algorithm and our algorithm. Middle: $\sqrt{n}$ divided by the number of phases executed by the HK algorithm. Right: $n^{1/3}$ divided by the number of phases of our algorithm. Note that the first two plots also appear in Figure 2.

matching algorithm tries several shifts of a grid in order to minimize the value of $\omega$. However, in practice, taking a single random shift of the grid is sufficient for finding a small number of boundary vertices in expectation. Recall that, given this shift, the corresponding $\delta$-disc graph can be created in $\tilde{O}(n)$ time, which is asymptotically far less than the overall complexity of our algorithm. An additional difference between the HK algorithm and our algorithm is that the our algorithm begins with a preprocessing step, which executes the HK algorithm within each piece. However, this step takes asymptotically less time than the subsequent phases of the algorithm (see Figure 4). Additionally, our experiments found that the total time taken by the edge weight assignment and preprocessing step was insignificant in comparison to the total time taken by our algorithm. The third difference between the HK algorithm and our algorithm is the fact that the our algorithm executes a $0/1$ BFS, which is slightly different from the version of BFS used by the HK algorithm. Within each phase, the $0/1$ BFS can be implemented to run in $O(m)$ time without any significant difference in performance in comparison to the HK algorithm's BFS.

The fourth and final difference between the two algorithms is that our algorithm finds augmenting paths during Stage 2 that are not necessary vertex-disjoint. While the first three differences do not have any significant direct impact on the running time, this final difference is of primary importance. There are two main effects of finding non-disjoint augmenting paths. First, more augmenting paths can be found during each phase, which decreases the number of phases that need to be run, decreasing the relative running time of our algorithm. However, a second effect is that edges may be revisited multiple times during Stage 2 of the same phase of our algorithm, increasing the relative running time of our algorithm.

Next, we present results that illustrate this dynamic in practice. First, we find that our algorithm executes significantly fewer phases than the HK algorithm. Also, as the guess of the bottleneck distance approaches the actual bottleneck distance, both algorithms seems to exhibit close to their worst case theoretical upper bounds of $O(\sqrt{n})$ and $O(n^{1/3})$ respectively for their number of phases (see Figure 3).

Finally, to better understand the running time of our algorithm, we divide the edge visits into three groups - the edges visits that occur during the preprocessing step, the edge visits where an edge is visited for the first time during a phase, and the times an edge is revisited during Stage 2 of a particular phase. These results are given in Figure 4. Note that, from these results, we can conclude that Stage 2 accounts for the majority of the running time, and most of the time taken during Stage 2 is due to revisits of edges.

## H   Details of dynamic nearest neighbor data structures

In this section, we provide additional details as to how the HK algorithm and our algorithm can be implemented using dynamic nearest neighbor (DNN) data structures. Let $G(V = A \cup B, E)$ be an arbitrary undirected graph where every edge $(u, v) \in E$ has a weight $w(u, v)$ that is either 0 or 1. A DNN structure $\mathcal{D}$ on the graph $G$ stores a subset $A' \subseteq A$ of vertices. Given an arbitrary query vertex $b \in B$, the data structure $\mathcal{D}$ returns a vertex $a \in A'$ that minimizes the weight of the edge $(b, a)$ (i.e., returning weight 0 edges before weight 1 edges). If there is no edge directed from $b$ to any $a \in A'$, then the query returns NULL. The DNN $\mathcal{D}$ supports queries in $\Phi(n)$ time. Furthermore, $\mathcal{D}$ allows an

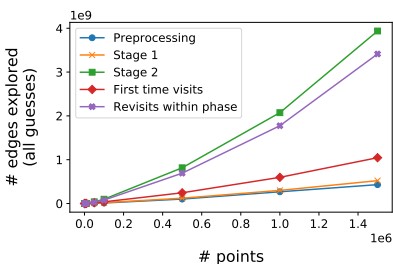

Figure 4: A breakdown of the edge visit counts of our algorithm into categories. First time visits occur whenever an edge is visited for the first time during a particular stage (either Stage 1 or Stage 2) of a phase; all subsequent visits to that edge during the same stage are revisits. Revisits only occur during Stage 2.

arbitrary vertex $a \in A \setminus A'$ to be inserted into $A'$ in $\Phi(n)$ time, and an arbitrary vertex $a \in A'$ to be removed from $A'$ in $\Phi(n)$ time.

In section H.1, we explain how the HK algorithm can be implemented, using a DNN $\mathcal{D}$, to run in $O(n^{3/2}\Phi(n))$ time. In section H.2, we explain how our algorithm can be implemented to run in $O(\Phi(n)(n\sqrt{r} + n\sqrt{\omega} + r\omega \log n))$ time using a DNN $\mathcal{D}$. In section H.3, we explain how such a DNN can be implemented for $\delta$-disc graphs.

### H.1    HK algorithm using DNN

In this section, we describe how the HK algorithm can be implemented to run in $O(n^{3/2}\Phi(n))$ time using a DNN data structure $\mathcal{D}$. Note that the HK algorithm does not use the edge weights $w(\cdot, \cdot)$; instead, queries to $\mathcal{D}$ are only necessary in order to identify *an arbitrary* edge of $E$ from a query vertex $b \in B$ to a vertex $a \in A'$.

**Stage 1:**    Next, we describe how the HK algorithm can use $\mathcal{D}$ to support BFS on the residual graph $G_M$ in $O(n\Phi(n))$ time. The purpose of this BFS is to identify, for any vertex $v \in A \cup B$, the minimum distance $\ell_v$ (in terms of total number of edges) from $s$ to $v$ in $G_M$. During the BFS, let $S$ be the set of vertices in $G_M$ that have been explored by the BFS, and let $T = V \setminus S$ be the set of vertices that have not yet been reached by the BFS. Throughout the algorithm, the set $A'$ maintained by the DNN data structure $\mathcal{D}$ will be equal to $A \cap T$. Initially, $S = \{s\}$, $T = A \cup B$, and $A' = A$.

Recall that a standard BFS can be implemented using a queue $Q$ that contains vertices of $V$. In $O(1)$ time, a single vertex can be added to the tail of $Q$ or removed from the head of $Q$. The queue $Q$ contains vertices that are *partially explored* and any vertex that is removed from $Q$ is *fully explored*. During each iteration of the BFS, let $v$ be the vertex at the head of $Q$. If $v \in A$, then there is at most one matching edge $(v, b)$ directed from $v$ to some $b \in B$. If $b \in T$, then the algorithm sets $\ell_b \leftarrow \ell_v + 1$, removes $b$ from $T$, adds $b$ to $S$, and adds $b$ to $Q$. Otherwise, if $v \in B$, then there could be many edges outgoing from $v$ in $G_M$. We are only interested in edges directed from $v$ to some $a \in A \cap T$. Conveniently, such a vertex $a$ can be identified in $O(\Phi(n))$ time by querying $\mathcal{D}$. Given such a vertex $a$, the algorithm sets $\ell_a \leftarrow \ell_v + 1$, removes $a$ from $T$, removes $a$ from $A'$, adds $a$ to $S$, and adds $a$ to $Q$. If the query to $\mathcal{D}$ on $v$ returns NULL, then all applicable neighbors of $v$ have been explored, and the algorithm removes $v$ from $Q$.

**Stage 2:**    Recall that Stage 2 of the HK algorithm conducts multiple DFS searches in order to find augmenting paths in the admissible graph $\mathcal{A}$, which contains all edges $(u, v)$ in $G_M$ with $\ell_v = \ell_u + 1$. It may be prohibitively expensive to construct $\mathcal{A}$ explicitly, as it could have $\Theta(m)$ edges. Instead, the algorithm will make use of multiple DNN data structures in order to efficiently identify the next edge to explore during a DFS. Recall that $\ell = \min_{a \in A_F}$ is the minimum distance from $s$ to any free vertex of $A_F$. For each even-valued $i \in [1, \ell]$ the algorithm maintains a DNN data structure $\mathcal{D}_i$. Each such data structure $\mathcal{D}_i$ contains a set $A'_i \subseteq A$, which initially contains all vertices $a \in A$ with $\ell_a = i$. During Stage 2, if a vertex $a \in A$ is reached during the DFS, the algorithm removes $a$ from $A'_{\ell_a}$. When the DFS is exploring neighbors of some $b \in B$, it will query $\mathcal{D}_{\ell_b+1}$ on $b$. If the query returns a

vertex $a \in A'_i$, then $(b, a)$ is an admissible edge, and $a$ has not previously been reached by a DFS. The algorithm removes $a$ from $A'_{\ell_a}$ and continues the DFS from $a$. If the query to $\mathcal{D}_{\ell_b+1}$ returns NULL, then no unexplored neighbors of $b$ remain, and the DFS backtracks from $b$. Note that, like during the BFS, any vertex $a \in A$ has at most one outgoing edge, which is a matching edge. As a result, the neighbors of any vertex $a \in A$ can be considered explicitly, without need of a DNN data structure.

**Efficiency:** To bound the running time of BFS using $\mathcal{D}$, observe that each vertex of $A$ is removed from $A'$ at most once. Furthermore, whenever a query is made to $\mathcal{D}$, either it returns NULL (which only occurs at most once per vertex of $B$) or else a vertex is removed from $A'$. Therefore, just $O(n)$ operations are required on $\mathcal{D}$. Similarly, during Stage 2, each vertex of $A$ exists in at most one set $A'_i$, and is removed from this set at most once. Therefore, Stage 2 also requires only $O(n)$ operations on DNN data structures, each of which take $\Phi(n)$ time. As a result, a single phase of the HK algorithm can be implemented to take $O(n\Phi(n))$ time. Over all $O(\sqrt{n})$ phases of the algorithm, the total time taken is $O(n^{3/2}\Phi(n))$.

## H.2 Our algorithm using DNN

In this section, given a DNN data structure on $G$, we explain how our algorithm can be made to run in $O(\Phi(n)(n\sqrt{r} + n\sqrt{\omega} + r\omega \log n))$ time. Most of the details are similar to the description of the HK algorithm using a DNN data structure, so we will focus on highlighting the key differences.

**Stage 1:** For Stage 1, there is one key difference between the HK algorithm and our algorithm. When selecting the next edge directed from $S$ to $T$ to explore, the HK algorithm ignores weights while our algorithm must select an edge with weight 0 if one is present. In order to support this in our algorithm, one option is to maintain two queues $Q_0$ and $Q_1$, each of which contain vertices of $A \cup B$. During each iteration of the BFS, the algorithm identifies an edge from $S$ to $T$ by checking $Q_0$ first, and only checking $Q_1$ if $Q_0$ is empty. If $Q_0$ is not empty, let $u \in S$ be the head of $Q_0$. If $u \in A$, then the neighbors of $u$ can be processed without using the DNN structure, so we consider when $u \in B$. Then an edge of weight 0 directed from $u \in S \cap B$ to $v \in T \cap A$ can be identified, if such an edge exists, by querying $\mathcal{D}$. If the result is an edge with weight 0, then the algorithm sets $\ell_v \leftarrow \ell_u$, adds $v$ to $Q_0$, adds $v$ to $S$, and removes $v$ from $T$. Otherwise, $u$ is removed from $Q_0$ and added to $Q_1$, and the algorithm continues to the next iteration. If $Q_0$ is empty, then the vertex at the head of $Q_1$ is processed instead. In a similar fashion, an edge of weight 1 directed from $u \in S \cap B$ to some $v \in T \cap A$ can be identified, if such an edge exists, by querying $\mathcal{D}$. If the result is an edge with weight 1, then the algorithm sets $\ell_v \leftarrow \ell_u + 1$, adds $v$ to $S$, and removes $v$ from $T$. Otherwise, the algorithm removes $u$ from $Q_1$ and continues to the next iteration.

**Stage 2:** The second stage of our algorithm has two key differences from the second stage of the HK algorithm that affect the use of the DNN structures: (i) Our algorithm can contain edges of weight 0 in the admissible graph $\mathcal{A}$. (ii) Unlike the HK algorithm, which deletes all visited vertices at the end of a single DFS, our algorithm only deletes the edges that did not participate in an affected piece.

To address (i), instead of making a data structure $\mathcal{D}_i$ for only the even values of $i$, our algorithm creates a DNN data structure $\mathcal{D}_i$ for every $i \in [1, \ell]$. When exploring a vertex $b \in B$ during the DFS, in order to identify an admissible edge outgoing from $b$, it is sufficient to query $\mathcal{D}_{\ell_b}$ as well as $\mathcal{D}_{\ell_b+1}$. If the query to $\mathcal{D}_{\ell_b}$ returns either NULL or a vertex $a$ with $w(b, a) = 1$ then there are no admissible edges from $b$ to a vertex of $\mathcal{D}_{\ell_b}$. Furthermore, if the query to $\mathcal{D}_{\ell_b+1}$ returns a vertex $a$, then $w(b, a)$ must be 1; otherwise, if $w(b, a) = 0$, then $a$ could have been reached via $(b, a)$ during Stage 1, and should have been included in $\mathcal{D}_{\ell_b}$ instead of $\mathcal{D}_{\ell_b+1}$.

For addressing (ii), first note that, if an edge directed from $u$ to $v$ is deleted during our algorithm (considering momentarily the version of our algorithm that does not use DNN structures), then $u$ has been visited by the previous DFS, but the previous DFS backtracked from $u$; otherwise, $u$ would participate in the augmenting path found by the DFS, contradicting the assumption that $(u, v)$ was deleted. Since the previous DFS backtracked from $u$, all remaining admissible outgoing edges from $u$ to a vertex of $T$ have been explored. Furthermore, since $u$ is not in an affected piece, no edge incident on $u$ is in an affected piece. Therefore, all admissible edges outgoing from $u$ are deleted, implying that $u$ can be deleted as well. We conclude that, instead of deleting all the visited edges that do not participate in affected pieces after a DFS, it is permissible to delete all of the visited *vertices*

that do not participate in affected pieces. This observation is applicable to an implementation of the our algorithm with or without the use of DNN structures. Note, however, that there could be some edges with weight 1 that were explored during the DFS for which both endpoints were in affected pieces. Since edges with weight 1 are not inside affected pieces, such edges should be deleted. In order to avoid reexploring these edges, whenever the algorithm backtracks from a vertex $b \in B$, that vertex should, for the remainder of Stage 2, only query $\mathcal{D}_{\ell_b}$; it should no longer use $\mathcal{D}_{\ell_b+1}$.

Given this information, we can now address (ii). Recall that, whenever the DFS reaches a vertex $a \in A$, that vertex is removed from $A'_{\ell_a}$. For the purposes of querying the DNN, this corresponds to deleting the vertex $a$. By the end of the DFS, any vertex of $A$ that was explored has been removed from its corresponding DNN structure. Thus, in order allow vertices in affected pieces to be explored again, it suffices to re-insert the deleted vertices from affected pieces back into their corresponding DNN structures. Specifically, as the DFS progresses, the algorithm maintains a set $\mathcal{X}$ of vertices of $A$ that have been reached. Whenever the DFS finishes, the algorithm will, for every $a \in \mathcal{X}$, insert $a$ back into $A'_{\ell_a}$ if $a$ belongs to an affected piece. All other vertices of $\mathcal{X}$ are deleted for the remainder of the phase.

**Efficiency:** For Stage 1, each vertex of $A$ is removed from $A'$ at most once, and each query to $\mathcal{D}$ results in either (i) the deletion of a vertex from $A'$, (ii) the removal of a vertex from $Q_0$, or (iii) the removal of a vertex from $Q_1$. Sine each vertex is inserted into $Q_0$ and $Q_1$ at most once each, the number of operations on $\mathcal{D}$ is $O(n)$.

Next, we bound the time taken by Stage 2. First note that, when identifying an admissible outgoing edge from a vertex $b \in B$, our algorithm makes two queries, one to $\mathcal{D}_{\ell_b}$ and one to $\mathcal{D}_{\ell_b+1}$, as opposed to the HK algorithm, which makes a single query. This difference does not asymptotically affect the running time. Each such pair of queries causes at least one of the following to happen: (i) A vertex is removed from $\mathcal{D}_{\ell_b}$, (ii) a vertex is removed from $\mathcal{D}_{\ell_b+1}$, or (iii) the DFS backtracks from the vertex $b$. Also observe that the number of deletions from each of the DNN structures is upper bounded by its total number of insertions.

During the second stage of a single phase, each operation on the DNN structures can be attributed to a single vertex's visitation, where each vertex visitation is responsible for $O(1)$ operations. The total number of vertex visits during Stage 2 of a single phase is $O(n)$ plus the total number of revisits. Recall that Lahn and Raghvendra bound the total number of affected pieces as $\omega \log \omega$ [28]. Whenever a piece is an affected piece with respect to an augmenting path, at most $O(r)$ vertices are revisited during that phase. Therefore, the total number of revisits during the entire algorithm is $O(\omega r \log \omega)$. With each vertex visit taking $\Phi(n)$ time, the total time taken by all $O(\sqrt{\omega})$ phases of our algorithm is $O(\Phi(n)(n\sqrt{\omega} + r\omega \log n))$. However, we must also consider the time taken by the preprocessing step. The preprocessing step can apply the HK algorithm, using the DNN structures as described in Section H.1. Since each piece has $O(r)$ vertices, the time taken by each piece is $O(r^{3/2}\Phi(r))$. Over all pieces, the total time taken for preprocessing is $O(n\sqrt{r}\Phi(n))$. Combining the time taken for the preprocessing and the phases gives a total running time of $O(\Phi(n)(n\sqrt{r} + n\sqrt{\omega} + r\omega \log n))$.

### H.3 DNN for $\delta$-disc graphs

The algorithms of sections H.1 and H.2 assume the existence of a dynamic nearest neighbor data structure for graphs with edge weights of either 0 or 1 that maintains a set $A' \subseteq A$ and, for any query vertex $b \in B$, returns a vertex $a \in A'$ that minimizes $w(b, a)$, or else NULL if no edge exists between $b$ and any vertex of $A'$. In this section, we describe how to support such a data structure in the context of $\delta$-disc matching, considering point sets $A, B \subset \mathbb{R}^2$. It is known how to support a similar dynamic nearest neighbor structure when every edge $(a, b) \in A \times B$ exists and has a weight equal to the Euclidean distance between its endpoints. This *Euclidean* DNN supports insertions, deletions, and queries in $\Phi'(n) = \log^{O(1)}(n)$ time. We describe how this Euclidean DNN can be used to support a $0/1$ edge-weighted DNN in the context of the $\delta$-disc matching algorithm described in Section 3.1.

Recall that, in the $\delta$-disc graph matching algorithm, any edge whose endpoints are in the same piece are assigned a weight of 0 and all other edges, whose endpoints lie in different pieces, are assigned a weight of 1. To implement the operations of $\mathcal{D}$, we construct a *global* Euclidean DNN structure $\mathcal{D}'$ on the vertices of the entire graph, as well as a *local* Euclidean DNN structure $\mathcal{D}'_j$ on the vertices within each piece $\mathcal{P}_j, 1 \leq j \leq t$. The global data structure $\mathcal{D}'$ maintains the same subset $A'$ as $\mathcal{D}$, and

the data structure $\mathcal{D}'_j$ for each piece $\mathcal{P}_j$ maintains the subset $A' \cap \mathcal{V}_j$, supporting queries on vertices $B \cap \mathcal{V}_j$ within the piece.

Given this setup, $\mathcal{D}$ supports queries on a vertex $b \in B$ with respect to the set $A'$ as follows: Let $\mathcal{P}_j$ be the piece that contains $b$. First, the algorithm will query $\mathcal{D}'_j$ in order to obtain the point $a \in A' \cap \mathcal{V}_j$ that is closest to $b$ within the piece. If the resulting Euclidean distance $d(a, b)$ between $a$ and $b$ is at most $\delta$, then $(a, b)$ exists in the $\delta$-disc graph, and has a weight $w(a, b) = 0$. The data structure returns $a$. Otherwise, there is no point $a \in A' \cap \mathcal{V}_j$ that is within a distance of $\delta$ from $b$, and we can conclude that there are no weight $0$ edges incident on $b$ with respect to $A'$. In this case, the algorithm proceeds to query the global data structure $\mathcal{D}'$ in order to obtain a weight $1$ edge incident on $b$. If $\mathcal{D}'$ returns a point $a \in A'$ that is within a distance $\delta$ from the query $b$, then $w(a, b) = 1$, and $\mathcal{D}$ returns $a$. Otherwise, there are no edges incident on $B$ with respect to $A'$, and $\mathcal{D}$ returns NULL. To insert a point $a \in \mathcal{V}_j$ into the set $A'$, the data structure $\mathcal{D}$ can simply insert $a$ into both $\mathcal{D}'$ and $\mathcal{D}'_j$. Similarly, $a$ can be removed from $A'$ by removing $a$ from both $\mathcal{D}'$ and $\mathcal{D}'_j$. Since each query, insertion, and deletion operation on $\mathcal{D}$ requires only $O(1)$ operations on corresponding Euclidean DNN structures, each operation on $\mathcal{D}$ can be supported in $\Phi(n) = O(\Phi'(n)) = \log^{O(1)}(n)$ time.