# OpenReview forum: "A Faster Maximum Cardinality Matching Algorithm with Applications in Machine Learning"
_NeurIPS.cc/2021/Conference — NeurIPS 2021 Poster_

### Official Review · Reviewer_HbxE · 2021-07-16

**Rating:** 7
**Confidence:** 4

**Summary:**

This paper gives a fast and practical algorithm for computing approximate max cardinality matching in unit disc planar graphs. It's based on a recently improved bound that utilizes the (2-level) partition tree in more redundant manners. The implemented algorithm was tested on moderate datasets of around 10^6 vertices, and a performance gain by a factor of 4 or so was demonstrated.

**Limitations And Societal Impact:**

The experiments feel lacking to me for several reasons:
* the runtime of ~ 1000s on 12 cores for 10^6 points is only an order of magnitude off from generating the entire graph of all pairs of 10^6 points (on a single core one can check about 10^9 such pairs in a second). So the gains over even a naive n^2 algorithm feels minimal.
* the factor 4 gain over the HK baseline feels marginal, and doesn't show a clear asymptotic improvement

**Main Review:**

On one hand, the LR algorithm is solid progress on a well known theoretical problem that also has direct connections with geometric optimal transport theory. On the other hand, the empirical studies only compares with their own baseline, and on random data that may or may not be representative of important cases of optimal transport. I also feel that such studies should be done in conjunction with various regularizers used in OT algorithms, as the gradients from those methods are what truly makes OT routines powerful.

While the gains of this method are not outstanding, they are definitely noticeable. I also thank the authors for clarifying several aspects of their setup, as well as runtime overheads that I didn't pick out on the first read. On the more theoretical side, I do believe extensions of these ideas can be quite important for OT based algorithms. So I feel the ideas here have potential, and this paper should be of interest to many working on OT algorithms

**Time Spent Reviewing:**

2

---

> ### Author Response · Authors · 2021-08-09
> **Response to review by HbXE**
>
> We would like to thank the reviewer for their comments and the take the opportunity to clear some confusion.
>
> Although our experiments were run on a machine with 12 cores, each experiment used only one core. We used multi-threading only to conduct multiple independent experiments in parallel, with each test in its own designated thread. Thus, the results reflect execution time of each experiment on a single core. Also, note that, in the chart that depicts actual running times, the values indicated give the total time taken by the HK and LR algorithms over all guesses of bottleneck distance $\delta$. The number of guesses made was roughly $30$ for $n=1,000,000$, resulting in high precision. For each guess, we execute LR-algorithm and HK-algorithm from scratch. Thus, our execution times can be seen as 30 executions of LR (resp. HK algorithm). This is why the total execution time ends up in 1000s of seconds. The execution time for the final guess (which is the slowest one) is approximately 50 seconds (for n=1,000,000).
>
> With regards to comparing with other optimal transport methods, we highlight the remark in the book (Computational Optimal Transport) by Peyre and Cuturi, (page 23, Remark 2.20) "In contrast to $p < \infty$, this ($p=\infty$) is a non-convex optimization problem which is difficult to solve numerically and to study theoretically." Indeed, as far as we know, the existing optimal transport approximation algorithms do not extend to $p=\infty$ or the bottleneck distance and the only known alternate solution is the HK-based algorithm, which we compare against.
>
> We do believe that the methods presented in our paper should prove beneficial to real geometric sets as well. However, to benefit from our algorithm one may have to carefully set $0/1$ weights on the edges of the $\delta$-disc graph by possibly using adaptive grids (based on densities). Any such study will require extensive experimental evaluation of various weight assignment strategies. We consider such a study to be an important next step, but out of scope for this paper.

---

### Official Review · Reviewer_bfYJ · 2021-07-16

**Rating:** 6
**Confidence:** 3

**Summary:**

This work applies a recent approach to maximum cardinality matching due to to Lahn and Raghvendra to a special case relevant to the machine learning community. Namely, they analyze the performance of this approach on delta-disc graphs in the service of computing popular distance metrics. Comparisons are made to the classic Hopcroft-Karp approach with empirical validation.

**Main Review:**

Overall, the paper provides good insights, but could be stronger. I am more familiar with related work in maximum cardinality matching and computational geometry than the distances addressed here. However, the paper makes some nice connections and appears to be a valuable contribution. In terms of originality, this is mainly a novel application and analysis of existing techniques.

The main weaknesses are the limited experiments and lack of clarity around contributions. Given that experiments form a significant part of the validation, I would like to see a bit more. Is there a good reason why these experiments should be sufficient? Or is there a reason for not including more experiments?

Regarding clarity and contributions, the paper understandably restates a lot of prior work, but it is sometimes unclear what is and isn’t novel to this work. For example, it isn’t clear from the presentation whether Theorem 1 is due to this paper or prior work without a very careful reading and knowledge of that prior work. After rereading various sections a few times, I think I have a correct understanding of what is new in the paper. However, I would appreciate if the authors could clarify for me which portions of the paper they believe are the key novel contributions. I also feel the paper needs to be revised for clarity around this issue.

There some important typos that should be corrected and all of the formal details should be carefully proofread:
Line 155: “free vertex a in B_F” should refer to “b” not “a”.
Line 174: “n” should be “\sqrt{n}”

**Time Spent Reviewing:**

5 hours

---

> ### Author Response · Authors · 2021-08-09
> **Response to Review by bfYJ**
>
> We would like to thank the reviewer for their comments. We begin by clarifying our contributions.
>
> Both the LR-algorithm and our algorithm will compute the same matching within the same asymptotic complexity. However, we replace the use of Dijkstra's shortest path procedure in the original LR-algorithm with a much simpler 0/1 Breadth First Search. This simplification is possible because we eliminate the need to maintain dual weights for points within the LR-algorithm. This simplification, we believe, is an important contribution of our paper. We will restate Theorem 1 to highlight this contribution better.
>
> Our second contribution makes connections between the LR-algorithm and  faster exact algorithms for several problems of interest to ML, i.e., bottleneck matching (Theorem 5), $\infty$-Wasserstein distance (Section 3.2), and, Levy-Prokhorov distance (Section 3.3). All these connections are made via the $\delta$-disc graph matching (Theorems 2, 3, and 4) and are novel contributions of this paper.
>
> All our results are validated via mathematical proofs. The main intention of the experiments were to show that our algorithm for bottleneck matching of random point sets were not only asymptotically better than HK-based algorithm, but the hidden constant within the execution time of our algorithm is not very large (thanks to the simplification of LR-algorithm) and so, our algorithm is competitive even for small values of $n$. This may not have been the case with the original LR-algorithm since it comes with substantial overheads such as, storage and update of dual weights, storage and update of slacks, use of priority queues and several additional comparisons that are required by the Dijkstra's algorithm.

---

> > ### Comment · Reviewer_bfYJ · 2021-08-30
> > **Acknowledgement of reading response**
> >
> > Thanks for addressing my questions about contributions. These clarifications were mostly in line with my understanding of the paper. I still believe the paper should be significantly revised/restructured for clarity. This is the biggest factor keeping this at a 6 score for me rather than 7.
> >
> > I appreciate the authors clarifying the intention of the experiments and largely agree that this is not the most important claim of the paper. However, I still feel that the paper would be strengthened with more convincing experiments supporting that claim.

---

### Official Review · Reviewer_oexc · 2021-07-16

**Rating:** 7
**Confidence:** 4

**Summary:**

This paper has the following contributions about algos for max matching on graphs with recursive separators:

1 simplifying the recent LR algo to not require dual weights

2 showing how this algo can find maximum matchings asymptotically faster than the classic (HK) algo for disc graphs with low "density"

3 showing how this algo for del-disc graphs can compute the bottleneck dist between point sets, and the Levy-Prokhorov distance efficiently, in low dim and when the measures are sufficiently spread out

4 experiments that show the proposed algo is faster than HK for 2 uniform point clouds in [0,1]^2

**Limitations And Societal Impact:**

ok

**Main Review:**

The contributions are solid and of interest to the TCS and comp geo communities. The observation that Levy-Prokhorov distance reduces to max matching in special graphs, while simple, is interesting to the stats and ML communities. The paper is also clearly written.

My main concern is that in order to have impact in the NeurIPS community, the authors should describe specific ML applications where this new algo could be useful. In particular, because this algo is designed for exact solutions in very low dimension (since needs recursive separators). Many ML applications of these distances are in high d, and also there are faster algos if approx solutions are satisfactory (which is the case if the optimization error is smaller than the error from modeling/sampling empirical measures/etc), eg a trivial O(n + poly(1/eps)) time alg for +-eps approx to W_inf and thus also bottleneck and Levy-Prokhorov by simply eps-gridding. Certainly high-precision algos in low dimension are very important - but a discussion should be added to clarify when exactly this might be used.

other comments:
- The algo does not seem parallelizable due to BFS/DFS. This is critical for ML scale. Is parallelization possible?
- The algo only seems to apply to Wasserstein distances W_p when p=infty. Correct? This should be clarified in the abstract. Especially because the Wasserstein distances of interest are typically W_1 and W_2.
- Consider stating Levy-Prokhorov reduction in its own lemma since its of general interest.
- Typos: L53 del->eps, L72 / -> *, L155: a->b, L181: V->E, L331 and Leighton, L406: of

**Time Spent Reviewing:**

5

---

> ### Author Response · Authors · 2021-08-09
> **Our response to the review by oexc**
>
>
> We thank the reviewer for their comments and suggestions and address the two important issues/topics about distances (Wasserstein (W) and Levy-Prokhorov (LP)) raised by them: a) application of our algorithm to approximating W and LP distances in low dimensions,  and, b) Importance of high-precision computation of W and LP distances between low-dimensional distributions.
>
> Both relative and additive approximations of both W and LP distances in low dimensions are based on an $\varepsilon$-gridding (for relative approximations, the value of $\varepsilon$ is chosen carefully; See Section 6 in Lahn and Raghvendra, JoCG 2021) of the point set followed by finding a maximum cardinality matching on a $\delta$-disc graph constructed on the centers of the grid cells. We can use our dual-free implementation of the LR-algorithm to obtain this maximum cardinality matching, leading to a substantially simpler relative approximation algorithm than described in the paper by Lahn and Raghvendra. We will add this extension of our result in the next version of our paper. As noted by the reviewer, for additive approximations, this same approach leads to a $O(n+\mathrm{poly}(1/\varepsilon))$ time algorithm. Using our algorithm, however, leads to a smaller exponent in $\mathrm{poly}(1/\varepsilon)$ as opposed to when the HK algorithm is used. Thus our simplified implementation of LR algorithm can be used to compute these distances for any precision and not just in the exact setting.
>
> Applications of high-precision computation of W and LP distances: We highlight three applications where exact computations are valuable. We will include a discussion on these in the next version of our paper.
>
>
> i) One can estimate W and LP distances between any two fixed-dimensional continuous distributions by simply computing the distances between samples drawn from the distribution. From the fact that both distances metrize weak convergence, for large enough samples we can get accurate distances. Faster high-precision algorithms play a crucial role in obtaining such estimates; see (Beugnot et al. UAI 2021, Bernton et al., Information and inference: A Journal of the IMA, 2019).
>
> ii) High dimensional Wasserstein distance is sometimes estimated via embedding them into several lower dimensional space and computing exact solutions in the lower dimensional space. For example, consider sliced Wasserstein distance; see for instance Kolouri et al, NeurIPS 2019.
>
> iii) In the emerging area of topological data analysis, high dimensional point clouds are characterized by two-dimensional point sets called persistence diagrams where each point represents the so-called 'birth' and 'death' time of a topological feature. Different high-dimensional point clouds can be compared by computing the bottleneck distance between the corresponding 2-d point sets (diagrams). More recently, other Wasserstein distances between persistence diagrams have also been considered. See for instance (Lacombe, Cuturi, Oudot, NeurIPS 2018, Vishwanath et al., NeurIPS 2020).
>
>
> One can exploit low-dimensional geometry to obtain parallel algorithms for Wasserstein distances, see for instance the paper by Andoni et al. STOC 2014 which provides a parallel algorithm for the $W_1$ Wasserstein distance. However, we acknowledge that theoretical improvements via parallel implementations is challenging and an open question. We will also clarify in the abstract that our approach is primarily for the $W_\infty$ and the Levy-Prokhorov distances.

---

> > ### Comment · Reviewer_oexc · 2021-08-31
> > **Response**
> >
> > Thanks to the authors for these clarifications. I have raised my score accordingly.

---

### Decision · Program_Chairs · 2021-09-27

**Decision:**

Accept (Poster)

**Comment:**

The reviewers consider the submission to have valuable contributions, and the ideas may help OT-based algorithms. There were some concerns about the presentation including:
1) the impact on the ML community is not clear;
2) the experiments are deficient: it is not clear that random point sets are in any way representative (e.g., they have a smaller value k, which may not be the case for a typical pointset).
The authors are encouraged to update the paper in accordance to the reviews/post-rebuttal comments.